# Dimethyl sulfide (DMS) climatologies, fluxes, and trends – Part A: Differences between seawater DMS estimations

**Sankirna D. Joge**[1,2], **Anoop S. Mahajan**[1], **Shrivardhan Hulswar**[1], **Christa A. Marandino**[3], **Martí Galí**[4,5], **Thomas G. Bell**[6], **and Rafel Simo**[4]

[1] TS1 , Indian Institute of Tropical Meteorology, Pune, India

[2] TS2 , Savitribai Phule Pune University, Pune, India

[3] Research Division 2-Biogeochemistry, GEOMAR Helmholtz Centre for Ocean Research Kiel, Kiel, Germany

[4] TS3 , Institut de Ciències del Mar (CSIC), Barcelona, Catalonia, Spain

[5] Barcelona Supercomputing Center (BSC), Barcelona, Spain

[6] Plymouth Marine Laboratory (PML), Plymouth, UK

**Correspondence:** Anoop S. Mahajan (anoop@tropmet.res.in)

**Abstract.** TS4 TS5 Dimethyl sulfide (DMS) is a naturally emitted trace gas that can affect the Earth's radiative budget by changing cloud albedo. Most atmospheric models that represent aerosol processes depend on regional or global distributions of seawater DMS concentrations and sea–air flux parameterizations to estimate its emissions. In this study, we analyse the differences between three estimations of seawater DMS, one of which is an observation-based interpolation method following Hulswar et al., (2022) (hereafter referred to as H22) and two of which are proxy-based parameterization methods following Galí et al. (2018) (hereafter referred to as G18) and Wang et al. (2020) (hereafter referred to as W20) CE1 . The interpolation-based method depends on the distribution of observations and the methods used to fill data between observations, while the parameterization-based methods rely on establishing a relationship between DMS and environmental parameters such as chlorophyll $a$, mixed-layer depth, nutrients, sea surface temperature, etc., which can then be used to predict DMS concentrations. On average, the interpolation-based methods show higher DMS values compared to the parameterization-based methods. Even though the interpolation method shows higher values than the parameterization-based methods, it fails to capture mesoscale variability. The regression-based parameterization method (G18) shows the lowest values compared to other estimations, especially in the Southern Ocean, which is the high-DMS region in austral summer. The parameterization-based methods suggest positive long-term trends in seawater DMS ($6.94 \pm 1.44$ % per decade for G18 and $3.53 \pm 0.53$ % per decade for W20). Since large differences, often more than 100 %, are observed between the different estimations of seawater DMS, the derived sea–air fluxes and, hence, the impact of DMS on the radiative budget are sensitive to the estimate used.

## 1 Introduction

Seawater dimethyl sulfide (DMS) is a major source of sulfate aerosols in the marine atmosphere (Bates and Quinn, 1997). It is a by-product of the phytoplankton life cycle and marine microbial food web interactions (Andreae and Crutzen, 1997; Simó, 2001). The produced DMS is either oxidized by photochemical reactions or metabolized by bacteria (Toole et al., 2003), and the rest is released into the atmosphere as gaseous DMS (Galí and Simó, 2015; Simó, 2001). In the atmosphere, DMS oxidizes to form sulfuric and methane sulfonic acid, eventually leading to aerosol formation and growth. These aerosols can act as cloud condensation nuclei (CCN), especially in environments removed from anthropogenic and continental influences (Andreae and Barnard, 1984; Korhonen et al., 2008). CCN contribute to the formation of clouds, increasing cloud albedo. Due to this, DMS emissions have the potential to decrease solar radiation at

the ocean surface, resulting in negative feedback (Vallina and Simó, 2007). This feedback cycle is referred to as the CLAW hypothesis (Charlson et al., 1987; Wang et al., 2018b). Past studies have shown that this feedback cycle is more complex than the original CLAW hypothesis (Quinn and Bates, 2011) However, it is undeniable that DMS affects the radiative budget on a global scale. For example, Fiddes et al. (2018) showed that the removal or enhancement of marine DMS can change the atmospheric radiative effect at the top of the atmosphere by 1.7 and $-1.4\,\mathrm{W\,m^{-2}}$, respectively. Mahajan et al. (2015b) showed that the difference between model simulations with and without DMS can result in an aerosol radiative forcing difference of $-0.179\,\mathrm{W\,m^{-2}}$ TS6, with the difference exceeding $20\,\mathrm{W\,m^{-2}}$ in the Southern Ocean. Hence, there is a need to understand the DMS cycle within the context of the uncertainties and biases of the climate models (Fossum et al., 2018; Fiddes et al., 2018).

The emission of DMS into the atmosphere is an important sea–air interaction process and determines the impact of seawater DMS on the global radiation budget (Stefels et al., 2007; Saint-Macary et al., 2022). In most global models, this flux is estimated as a product of the seawater DMS concentration and a parameterization of the sea–air flux transfer velocity (Liss, 1983; Johnson, 2010 TS7). Considering that seawater DMS concentration is an essential part of the flux calculation, its accurate estimation plays a crucial role in quantifying the impact of DMS on cloud formation. Regional and global distributions of seawater DMS concentrations are estimated using observation-based interpolation, process-level modelling, and parameterization-based approaches (Belviso et al., 2004b).

In the interpolation-based approach, the global seawater DMS distribution is estimated by interpolating and/or extrapolating all available in situ DMS observations. The first observation-based climatology was published by Kettle et al. (1999) and used only about 15 000 observations globally. Observations were segregated using static biogeochemical province boundaries defined by Longhurst et al. (1995) and were then interpolated across province boundaries and individual grid points. A similar approach was followed by Lana et al. (2011), although the number of data points used in this study had increased 3-fold (47 000 observations). Hulswar et al. (2022) recently presented an updated version, i.e. the third climatology, using an interpolation-based approach. This recent climatology was created with a $\sim$ 18-fold increase in observations (873 539 observations) and included important updates in the filtering and data unification process. They also included dynamically changing seasonal biogeochemical province boundaries (Reygondeau et al., 2013) to capture spatial and temporal changes in biogeochemistry, especially along the borders of provinces. The interpolation lengths for this climatology are based on observed DMS variability length scales (VLSs) (Royer et al., 2015; Manville et al., 2023), which produce more realistic geographical distributions.

In process-level models, the estimation of DMS is done using mathematical relationships at small scales between many biogeochemical and environmental parameters to define how DMS production and destruction occur. This method is complex due to the non-linear relationship between DMS; proxy parameters; and DMS's main precursor, dimethylsulfoniopropionate (DMSP). The biogeochemical cycle of nutrients and the spatiotemporal distribution of different plankton taxa play an important role, and these are modelled across the globe using a detailed biogeochemical model, which predicts the seawater DMS concentrations (Anderson et al., 2001; Wang et al., 2018a; Belviso et al., 2004b). These estimations are inherently linked to our understanding of the underlying processes controlling DMS production and loss and, hence, can be highly biased if these processes are not well described in the model (Galí et al., 2023). This method is also computationally expensive. The models based on this approach lead to DMS climatologies with resolutions which are dependent on their parent model but are usually of the order of $0.25° \times 0.25°$ and hence can include mesoscale dynamic changes.

Finally, in the parameterization-based approach, a parametric equation between DMS and/or DMSP and single or multiple variables (biogeochemical and/or environmental parameters) is defined through linear and/or multi-linear regression at a larger scale. This approach is simple to implement compared to process-level models and can work more efficiently than observation-based interpolation for capturing mesoscale changes and understanding trends (Belviso et al., 2004a). Initial attempts were made in the early 2000s, with Simó and Dachs (Simó and Dachs, 2002) using chlorophyll $a$ and mixed-layer depth (MLD) as proxies for predicting DMS. Later, Vallina and Simó (2007) additionally used surface irradiance as a predictor due to a strong relationship having been observed between DMS and the solar-radiation dose over the global surface ocean. A recent study derived the relationship between DMSP and satellite-based data of chlorophyll $a$, sea surface temperature (SST), particulate inorganic carbon (PIC), and MLD in both stratified and mixed water columns (Galí et al., 2015). Later, DMS values were estimated across the oceanic biomes as a function of estimated DMSP and the satellite-based data of photosynthetically available radiation (PAR) using a similar regression analysis (Galí et al., 2018). An upgrade to this method is using machine learning, such as an artificial neural network (ANN) (Wang et al., 2020) or Gaussian process regression (GPR) (Mansour et al., 2023) to create the parameterization. The climatology in these cases is created by training the machine learning algorithms in data-rich regions. While ANN is more expensive in terms of computation than regression analysis, it is less expensive than process-level models. The parameterization approach used within modelling simulations (Halloran et al., 2010) shows that the method is not applicable under all conditions for estimating DMS. The biggest disadvantage of the ANN method is that it requires a large

number of observations to train the model efficiently. ANN is composed of layers of interconnected nodes. These nodes are organized into three layers: input layer, hidden layer, and output layer. The hidden layer performs complex computations on the parameters obtained from the input layer and trains itself according to the parameters given to this layer. Once it is trained, the ANN becomes capable of predicting DMS values at a single node in the output layer. A series of sensitivity tests between DMS and the individual parameters need to be run to check whether a change in a single parameter gives a unidirectional response for the predicted DMS values (Wang et al., 2020).

We selected the latest interpolation-based estimation (Hulswar et al., 2022) and two parameterization-based DMS estimations (Galí et al., 2018; Wang et al., 2020) to study the relative differences in the absolute values of the estimations, as well as their geographical differences, and to compare the long-term trends.

## 2   Methods

In this study, we compare three seawater DMS estimations created (Figs. 1–3) using two methods: i.e. an interpolation-based climatology estimate following Hulswar et al. (2022), hereafter referred to as H22 (https://doi.org/10.17632/hyn62spny2.1), and two parameterization-based climatology estimates following Galí et al. (2018), hereafter referred to as G18 (https://doi.org/10.5281/zenodo.2558511), and Wang et al. (2020), hereafter referred to as W20 (https://doi.org/10.5281/zenodo.3833233) (Wang et al., 2020) CE2 . Figure S1 in the Supplement shows the in situ DMS used in G18, W20, and H22. As only monthly climatologies of DMS are available from G18 and W20 public data, the models from these two papers were re-run to get monthly estimates of DMS from the years 1998 to 2010 in order to calculate the trends of seawater DMS. The parameters used for W20 and G18 are sea surface temperature (SST), salinity, and nutrients (such as phosphate, nitrate, and silicate) from WOA 2018 (https://www.ncei.noaa.gov/access/world-ocean-atlas-2018/, last access: TS8 ) at a $1° \times 1°$ monthly resolution; MLD from MIMOC (https://www.pmel.noaa.gov/mimoc/, last access: TS9 ; $0.5° \times 0.5°$ and monthly resolution); and satellite-based variables from NASA SeaWiFS (https://oceancolor.gsfc.nasa.gov/l3/, last access: TS10 ; $9 \times 9$ km and monthly resolution) for chlorophyll $a$, PAR, euphotic depth, and PIC. Thus, DMS data for W20 and G18 were re-created at a $1°$ resolution, similarly to the resolution of H22. For this, input data were also regridded to $1°$ before running both the models. It should be noted that there is a limitation in using satellite data as proxy data. For example, if we consider the Southern Ocean, satellite data do not provide robust PAR values where sea ice is present, and the

general availability of satellite data is restricted south of $50°$ S in early spring and late autumn, which may bias the DMS climatology. In the case of G18, the $DMSP_t$ values were calculated based on the equations given by Galí et al. (2015), and then DMS monthly values were calculated using globally optimized coefficients for the parametric equation for $DMSP_t$-to-DMS conversion (Galí et al., 2018). For W20, we used the best combination that was determined by Wang et al. (2020) to train the model, resulting in an $R^2 = 0.66$.

The decadal trend for G18 and W20 is calculated using the bootstrap-resampling method (Geiger et al., 2002). Before applying the bootstrap method, the seasonal variation is removed from the DMS time series dataset. For this, the mean values of each month are calculated for the years 1998–2010 (due to availability of satellite data) and are then subtracted from the corresponding month of each year. This results in anomalies used for calculating the trend using the bootstrap-resampling method. The bootstrap method randomly selects samples ($n = 100$) with replacements from the entirety of the anomaly data, which are present from the year 1998 to 2010. These samples are fitted over a first-order polynomial, and the corresponding gradient (trend) and intercept are obtained for each sample set. After this, the mean trend ($B$) and corresponding standard deviation ($\sigma_B$), as well as the mean intercept and its corresponding standard deviation, are calculated. The $t_b$ value is obtained by taking the ratio of the mean trend ($B$) and its corresponding standard deviation ($\sigma_B$); i.e. $t_b = |B/\sigma_B|$. If the $t_b$ value is greater than 2 then the significance level of the trend and its intercept are considered to be better than 95 % (Weatherhead et al., 1998). This method has been used to calculate long-term trends in the past (Mahajan et al., 2015a).

## 3   Results and discussions

### 3.1   Differences between the DMS climatologies

The seasonal and geographical variation in the three seawater DMS climatologies is shown in Fig. 1a. Broadly, the seasonal variation is dominated by the available solar radiation, with peaks in the Northern Hemisphere during June–July–August (JJA) and peaks in the Southern Hemisphere during December–January–February (DJF). The maximum DMS values observed in the polar regions during their respective summers have been attributed to the melting of ice that releases nutrients at the time of maximal light availability (Hawkings et al., 2020; Becagli et al., 2016; Zhang et al., 2021; Park et al., 2019; Gourdal et al., 2018; Sørensen et al., 2017), which causes phytoplankton blooms in the Arctic and Antarctic coastal regions. Figure 1b shows the histogram of DMS concentrations. For all the climatologies, most pixels show DMS concentrations $< 3$ nM in the oligotrophic re-

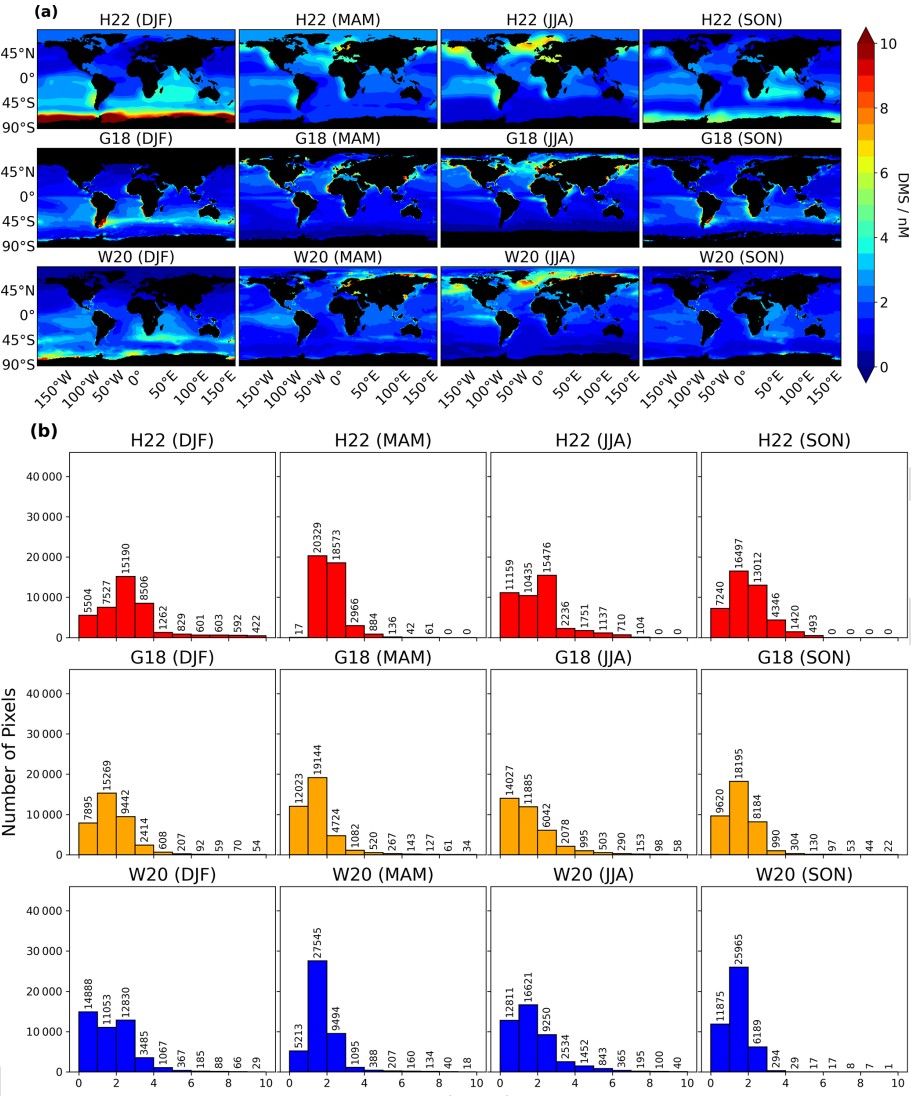

**Figure 1. (a)** Global seasonal climatologies of H22, G18, and W20 for austral summer (December–January–February (DJF)), spring (March–April–May (MAM)), boreal summer (June–July–August (JJA)), and autumn (September–October–November (SON)) seasons. For all the climatologies, most of the pixels show DMS concentrations of less than 3 nM in oligotrophic regions and higher concentration along coastal regions. **(b)** G18 and W20 captured DMS values of more than 8 nM, while H22 did not (except for the DJF season). H22 shows the highest number of pixels in the 3–4 nM range and more than 2000 pixels in total above 6 nM in the DJF season.

gions and higher concentrations along the coastal regions and regions with higher nutrient availability.

During the austral summer season (DJF), H22 shows a uniform increase in the Antarctic Circle and the Southern Ocean. By comparison, G18 does not show a peak in coastal Antarctica or the Southern Ocean, probably because of $1° \times 1°$ regridding. This is because re-gridding pixels results in lowering the peak values. There is poor agreement between all three climatologies in the Southern Hemisphere. A band of elevated DMS in the South Atlantic and Indian oceans centred around the 45° S latitude is seen in G18 (Fig. 1a). This is because chlorophyll-$a$ satellite data may be biased towards

coloured dissolved organic matter (CDOM) and detritus in the Argentinian basin (Astoreca et al., 2009; Hayashida et al., 2020; Bock et al., 2021). Thus, chlorophyll $a$ is considered to be a poor predictor by itself. This region is the transition between subtropical and subpolar waters and is also known for high abundances of DMS and the production of CE3 coccolithophores and dinoflagellates (Balch et al., 2016). However, H22 and W20 show a broader meridional spread (Fig. 1a). G18, which uses a regression-based parameterization and has coefficients that are sensitive to the PAR and, hence, to light-absorbing fractions such as CDOM and detritus, is most likely to be biased. W20 shows a distribu-

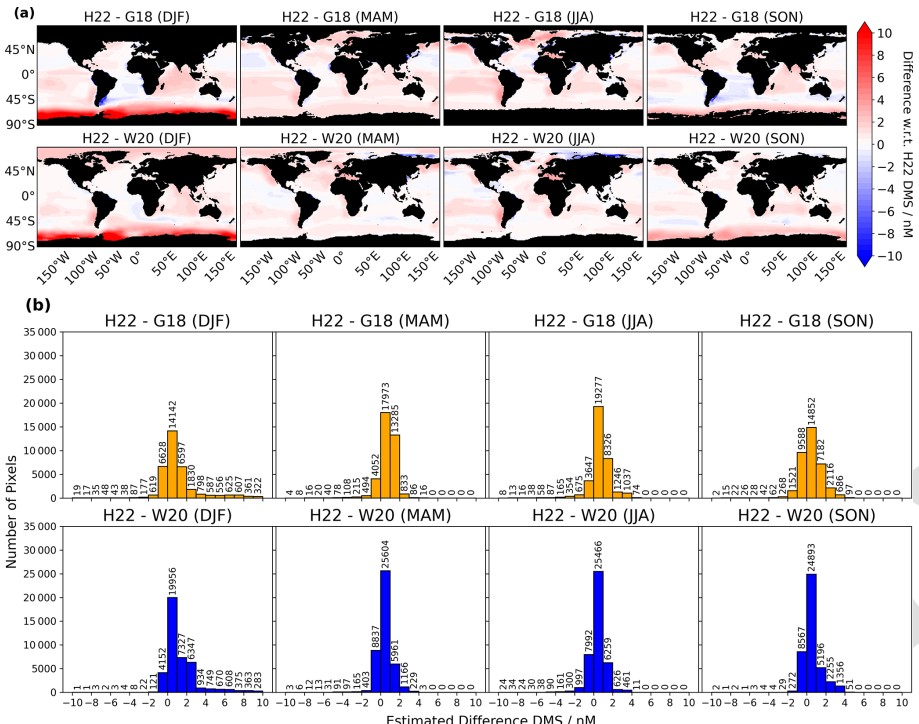

**Figure 2. (a)** Differences between the H22 climatology compared with G18 and W20. From all the seasons, the maximum difference between H22 and G18 is −14.74 nM during December–January–February (DJF) in the Argentinian basin and −29.03 nM for W20 during March–April–May (MAM) in the North Sea. **(b)** A histogram represents the total number of pixels for each difference bin. The differences between H22 and G18 or W20 are not exactly centred around zero, but the highest number of pixels show high values in the H22 estimation.

tion similar to that of H22, albeit with lower DMS values in most regions and higher values in the Ross Sea and Weddell Sea regions compared to the Indian sector of the Southern Ocean. The histogram distribution (Fig. 1b) also shows that H22 predicts higher values than the other two climatologies, with the highest number of pixels in the 3–4 nM range and more than 2000 pixels showing concentrations above 6 nM, while G18 has less than 300 pixels with concentrations above 6 nM (Fig. 1b). For G18, the pixels with higher concentrations are in the southern mid-latitude region or in coastal regions (Fig. 1a), while, for the other climatologies, most of these values are in the Southern Ocean and coastal Antarctica. G18 and W20 show fewer pixels with concentrations larger than 6 nM as compared to H22 (Fig. 1b).

A similar variation can be observed during the boreal summer season (JJA) in the Northern Hemisphere, where high concentrations of DMS are present in the Arctic Circle in all climatologies (Fig. 1a). The geographical distribution in the Northern Hemisphere during summer is similar for H22 and W20, with peaks being observed east of Greenland and off the coast of Alaska and with high values in the Arctic (Park et al., 2018). W20 shows peak values along the northern coastal regions of Russia in the Kara Sea and Laptev Sea regions CE4 compared to H22. G18 shows peak values in the Chorne Sea and Celtic Sea regions CE5. Both G18 and W20 show high lo-

cal peaks in terms of DMS concentration compared to H22. In terms of histogram distribution, G18 shows approximately 600 pixels with DMS concentrations of less than 6 nM, while W20 shows up to 800 pixels. For H22, this pixel count is approximately 800. It can also be observed that G18 and W20 captured DMS values of more than 8 nM, while, in H22, there were no values that high (Fig. 1b). The peak values observed during the boreal summer are lower than during the austral summer, with fewer pixels showing values above 6 nM for all the climatologies.

During boreal spring (March–April–May (MAM)) and autumn (September–October–November (SON)), there is a gradual increase in DMS concentrations in both the Northern Hemisphere and the Southern Hemisphere, as seen in Fig. 1a. The number of pixels with concentrations larger than 6 nM is low for all the climatologies (Fig. 1b). The H22 climatology shows higher values along the coastal-upwelling regions, such as South America's west coast and Africa's southwest coast (Fig. 1a), which was observed in previous studies. For example, the DMS concentration in the waters of the Peru upwelling region (Andreae, 1985; Riseman and DiTullio, 2004), the highest DMS concentration in the coastal-upwelling areas of the west coast of India (Shenoy and Kumar, 2007), North Africa, Angola, Peru, and the equatorial Pacific Ocean, is also observed (Kettle et al., 1999); Mau-

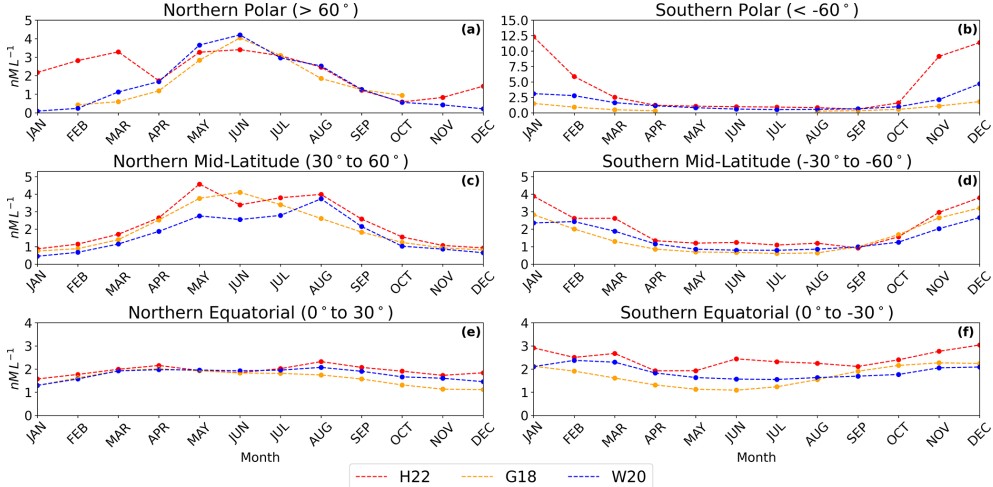

**Figure 3.** Latitudinal means for each month for all climatologies used in this study. Large differences are observed in the southern polar region between the interpolation-based and parameterization-based climatologies. G18 has the lowest values of the three in the southern polar region, while the estimates are close to those of W20 in the northern equatorial band.

ritanian upwelling is a hotspot for DMSP and thus DMS, which underlines coastal-upwelling regions as a local source for seawater DMS (Zindler et al., 2012). During SON, a peak is also seen in the Indian Ocean by all the climatologies due to the physical forcing generated by monsoon wind in the form of upwelling, which results in high biological production (Shenoy et al., 2002; Shenoy and Kumar, 2007), although G18 shows higher values in the Atlantic and Pacific too, which is missing in the other estimations.

The area-weighted global DMS means for the climatologies are 2.28 nM for H22, 1.69 nM for G18, and 1.75 nM for W20. Thus, the two parameterization-based estimations show lower global weighted-mean concentrations than the interpolation-based estimations. However, the parameterization-based estimations show higher peak values; for example, the maximum value during DJF is 18.67 nM in the Weddell Sea for H22, but this is higher at 18.94 nM off the coast of Chile in the South Pacific Ocean for G18 and at 23.64 nM in the Gulf of Mexico for W20. The maximum DMS during JJA for H22 is 7.29 nM in the Norwegian Sea, while, for G18, the peak is 15.84 nM in the Chorne Sea, and it is 46.23 nM in the Kara Sea for W20. This shows that, although globally averaged concentrations are higher in the interpolation-based method, the concentrations over individual pixels can be much larger for the parameterization-based approaches. The main reason for this is the bin-based averaging of observations done in the interpolation-based approach to remove very localized high values that would have a disproportionate weight in terms of regional and global averages. Due to this, no pixels higher than 8 nM are observed in H22 in MAM, JJA, and SON (Fig. 1b). Also, a sampling bias is inherent to the interpolation-based method, as discussed by Galí et al. (2018). Thus, the parameterization-based approaches have an advantage where they can capture large

point emissions during periods of high productivity. These high point emissions are likely to affect local and regional new particle formation on shorter timescales.

Figure 2a shows the absolute difference between H22 and the other two climatologies, while Fig. S2 shows the proportional differences. In the Southern Ocean, H22 predicts a higher value of DMS concentration, with larger positive differences compared to CE6 G18 and W20. In DJF, large negative differences can also be observed with G18 in the Argentinian Shelf region and in the coastal areas of Peru and Chile. Similarly, positive differences can also be seen in the JJA season, with some negative differences in the case of W20 in the Arctic Circle and with negative differences in the case of G18 along some coastal areas of the continents. The histogram of differences is centred around zero, showing that most pixels show a minor change, although large differences of $> 10$ nM are also seen in some pixels, especially during DJF. The differences between H22 and G18 or W20 (Fig. 2b) are not centred around zero, with most pixels showing higher values in the H22 estimation. Some pixels show a negative difference in the Arctic Ocean, southern Atlantic, and South Australian basin, mostly along high-productivity coastal regions. From all the seasons, the maximum difference between H22 and G18 is $-14.74$ nM during DJF in the Argentinian basin region and $-29.03$ nM for W20 during MAM in the Arctic Sea. Overall, G18 and W20 show a lower estimation than H22 in the Antarctic coastal area, but G18 shows higher values in the coastal regions of other continents, such as in South America in the coastal areas of Peru and Chile and in the Argentinian basin, as well as in the northern coastal regions of Russia.

The difference in the methods is driven by various factors. The sensitivity of the methods to certain parameters (or observation bias in the case of H22) is the primary driver. However, the main reason for this is the availability of high-

resolution observations across different regions and seasons and also the quality of the observations. In the future, more observations will help resolve some of these differences.

## 3.2 Latitudinal variations

The latitudinal variations of globally averaged seawater DMS climatologies for each month are shown in Fig. 3. We checked the variations according to six latitudinal regions, i.e. the northern polar region ($> 60°$ N) and the southern polar region ($> 60°$ S), the northern mid-latitude region (30 to $60°$ N) and the southern mid-latitude region (30 to $60°$ S), and the northern equatorial region (0 to $30°$ N) and the southern equatorial region (0 to $30°$ S). All the climatologies show a similar annual trend in all the regions, although considerable differences are observed in the polar regions.

In the northern polar region, H22 surprisingly shows a lower mean DMS value (1.73 nM) in April compared to in February and May (Fig. 3a). This is most likely due to faulty interpolation in H22, which indicates that observation-based interpolation methods can become biased if incorrect mapping is done. In the same region, a maximum mean value of 4.20 nM is observed in June, which is closer to that of G18 (4.04 nM) but higher than that of H22 (3.41 nM). H22 estimates high mean values in January, February, March, November, and December compared to G18 and W20. The W20 estimations closely match the interpolation-based estimations in the boreal summer months, and although both G18 and W20 follow the same pattern, lower values are observed in the winter months of DJF compared to H22. Considering the low sunlight during this period, the means suggest that the interpolation-based methods overestimate the DMS concentrations during winter, while W20 estimations seem to be more likely. This bias is most likely due to interpolation rather than a sampling bias.

Large differences are observed in the Southern Ocean between the interpolation-based and parameterization-based climatologies. With much increased data availability in the Southern Ocean owing to the high-frequency observations obtained using membrane inlet mass spectroscopy (MIMS), the updated DMS climatology in Jarníková and Tortell (2016), which was created using new high-frequency observation data in the Southern Ocean, shows higher concentrations in high latitudinal regions. The differences may reach over +10 nM in some regions, like in the Weddell Sea and in the waters around the Balleny Islands, while large underestimations of over −10 nM may appear in other regions, such as those of the Ross and the Bellingshausen seas. Although all the climatologies show higher values during the austral summer months, H22 (peak: 12.3 nM in January) shows higher values as compared to G18 (peak: 1.81 nM in December) and W20 (peak: 4.69 nM in December). G18 struggles to simulate accurate concentrations, suggesting that this method fails in southern polar regions (Fig. 3b). W20 shows an increase, although this is driven by higher concentrations in particular regions, such as in the Ross Sea, as compared to more generalized larger concentrations along the entire Antarctic coastline, as seen in H22 (Fig. 1a).

For the northern mid-latitude region, H22 shows values peaking at 4.57 nM (in May). W20 also shows an increase in the summer with values in the range of 2.75 nM (in May) to 3.73 nM (in August). G18 shows values ranging from 2.61 nM (in August) to 3.76 nM (in May) and peaking at 4.11 nM in June (Fig. 3c). In the southern mid-latitude region, which covers the Southern Pacific, Atlantic, and Indian oceans, H22 estimates a range from 2.96 nM in November to 3.88 nM in January. Estimates for G18 and W20 are similar, with peaks appearing in the austral summer months (between $\sim$ 2–3 nM; Fig. 3d). Although the means are similar for these two estimates, the geographical distribution is different; while G18 shows a band of increased DMS along the $45°$ S latitude, W20 shows increases along Africa and the Pacific.

The equatorial regions show the lowest mean concentrations of all the latitudinal regions. In the northern equatorial region, all the climatologies show a similar estimation, with values ranging between 1–2.5 nM. G18 shows lower values, especially from August to December. For the southern equatorial region, H22 peaks at 3.04 nM in December, while W20 and G18 show lower values; although, similarly to other regions, G18 gives the lowest values of the three from February to July for these latitudes.

## 3.3 Long-term trend

The long-term trends in DMS for G18 and W20 are shown in Fig. 4. High-temporal-resolution data are important for time series analysis to observe variations. For G18 and W20, the trend is calculated after removing the seasonal signal from the time series for data between the years 1998 and 2010. G18 (Fig. 4a) and W20 (Fig. 4b) show increasing trends of $6.94 \pm 1.44$ % per decade ($t_b = 4.82$) and $3.53 \pm 0.53$ % per decade ($t_b = 6.71$), respectively. This suggests an increase in globally averaged seawater DMS concentrations across the world's oceans. In the case of G18, the calculations are done using the globally optimized coefficients (Galí et al., 2018). If the same calculations are done using coefficients optimized for $> 45°$ N (Fig. S4) then the calculated trend is $7.20 \pm 1.90$ % per decade ($t_b = 3.80$). Thus, the trend in the W20 climatology is nearly 50 % lower than the trend observed by G18, probably due to the differences in the parameterization scheme and the sensitivity of coefficient values in relation to the different predictors in both methods. It should be noted that the radiative forcing of past and future DMS-driven aerosol formation is uncertain. The IPCC AR5 concluded that a negative feedback of $-0.02$ W m$^{-2}$ °C$^{-1}$ is expected (IPCC, 2014), with DMS emissions expected to increase with global warming. The AR6, in contrast, suggests that DMS emissions are expected to decrease, resulting in a positive feedback of 0.005 (0.0 to 0.01) W m$^{-2}$ °C$^{-1}$ (IPCC,

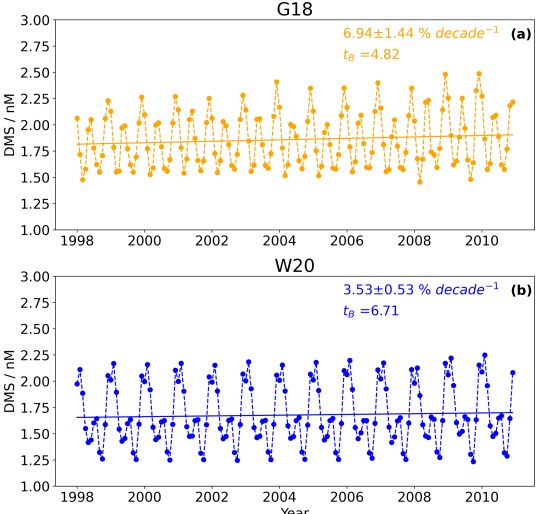

**Figure 4.** Interannual trends in all the seawater DMS concentrations for **(a)** G18 and **(b)** W20. The interannual trend is significant and positive. The trend is calculated using the bootstrap-resampling method.

2021) due to a decrease in ocean productivity. The results presented here show an increasing trend in the seawater DMS concentrations from the year 1998 to 2010 and suggest that more research is needed to understand the drivers of seawater DMS before an accurate estimation of its impacts in the future can be made. SeaWiFS satellite data are available only from the year 1998 to 2010, and the same limitation can be seen with other satellite products, which start from 2002 onwards CE7. Hence, there is a limitation in the past and future projection of DMS values due to the availability of satellite-based predictors for limited years. Even though an increasing trend is obtained in G18 and W20, this period is not sufficient to understand the long-term variability of the Earth system and the DMS response to it. In theory, this could be addressed using the machine learning code and proxies from climate model projections, although this has large uncertainties too.

### 3.4 Comparison with other climatologies

Over the last 2 decades, diagnostic or prognostic models – or models that are prognostic but use empirical modules to predict DMS – have been used to quantify the impact of DMS (Collins et al., 2011; Kloster et al., 2006; Six and Maier-Reimer, 2006; Vogt et al., 2010; Elliott, 2009). Hence, to compare the results from the observation-based interpolation method (H22), the regression-based parameterization (G18), and the machine-learning-based parameterization (W20), we choose only models that are either prognostic or diagnostic. These models are described as follows:

- Aumont et al. (2002) were the first to apply a process model parameterization for global DMS using chloro-phyll and community structure indices derived from a global biogeochemical model with a variable horizontal grid from 0.5 to 2°. This method estimated a weighted annual mean DMS of 1.70 nM.

- Chu et al. (2003) simulated the production and destruction of DMS by producing $DMSP_d$ through planktonic excretion of DMSP, which yields DMS through lysis. The DMS sinks included photolysis, bacterial consumption, and gas exchange at the air–sea interface, giving a high-resolution ($0.28° \times 0.28°$) estimate of DMS across the world's oceans. This prognostic model resulted in a weighted annual global mean DMS of 1.51 nM.

- The Centre National de Recherches Meteorologiques Earth System Model version 2 (CNRM-ESM2-1) (Séférian et al., 2019) computes DMS concentrations using the biogeochemical Pelagic Interactions Scheme for Carbon and Ecosystem Studies (PISCES) model (Aumont and Bopp, 2006). This includes the processing of DMSP to DMS and phytoplankton functional groups with the destruction of DMS via bacterial decomposition, photolysis, and ventilation. The model computes a weighted annual global mean DMS of 1.98 nM.

- The Norwegian Earth System Model, version 2, with Low-resolution atmosphere–land and Medium-resolution ocean sea ice CE9 (NorESM2-LM) (Seland et al., 2020) does not describe the conversion of DMSP to DMS, like in PISCES; instead, it directly computes DMS as a function of temperature, resulting in a weighted annual global mean DMS of 1.98 nM.

- The Model for Interdisciplinary Research On Climate, Earth System version 2 for Long-term simulations CE10 (MIROC-ES2L) (Hajima et al., 2020) computes the seawater DMS concentrations using a modified parameterization of Simó and Dachs (Simó and Dachs, 2002) that uses MLD and chlorophyll in two regimes (open ocean and shallow mixed water), depending on the chlorophyll-to-MLD ratio. This results in a weighted annual global mean DMS of 1.77 nM.

- The United Kingdom Earth System Model, version 1, with Low resolution for both atmosphere–land and ocean sea ice CE11 (UKESM1-0-LL) (Sellar et al., 2019) is used to compute the DMS concentration within the biogeochemical Model of Ecosystem Dynamics, nutrient Utilization, Sequestration and Acidification (MEDUSA) (Yool et al., 2013) based on the parameterization given by Anderson (Anderson et al., 2001), in which DMS concentrations depend on a logarithmic function of light, chlorophyll, and nutrients. The parameterization used in this model results in a weighted annual global mean DMS of 1.78 nM.

**Table 1.** Summary of the different methods and the respective area-weighted global annual mean DMS values. CE8

| Climatology/ model | Area-weighted global DMS mean (nM) | Characteristics of DMS scheme | Reference |
|---|---|---|---|
| H22 | 2.28 | Interpolation | Hulswar et al. (2022) |
| W20 | 1.75 | Machine learning-based parameterization | Wang et al. (2020) |
| G18 | 1.69 | Simple regression-based parameterization | Galí et al. (2018) |
| Au02 | 1.70 | Process model parameterization | Aumont (2002) |
| Chu03 | 1.51 | Prognostic model | Chu et al. (2003) |
| CNRM-ESM2-1 | 1.98 | Prognostic model | Séférian et al. (2019) |
| NorESM2-LM | 1.98 | Prognostic model | Seland et al. (2020) |
| MIROC-ES2L | 1.77 | Diagnostic model | Hajima et al. (2020) |
| UKESM1-0-LL | 1.78 | Diagnostic model | Sellar et al. (2019) |

CNRM-ESM2-1 and NorESM2-LM are prognostic models that include marine biota that include sinks and sources of DMS and/or DMSP, while MIROC-ES2L and UKESM1-0-LL are diagnostic models that use empirical parameterizations based on chlorophyll and other parameters (Bock et al., 2021). From Table 1, it can be observed that the global area-weighted annual mean DMS range (1.51–1.98 nM) of all these models is close to the weighted annual mean DMS of W20 (1.75 nM) and G18 (1.69 nM). The area-weighted global annual means computed by the interpolation-based approach (H22) is higher (2.28 nM) than those of these models. Most models follow the parameterization approach in order to define the production and destruction processes of DMS with environmental or biogeochemical parameters, which depend on our understanding of the underlying processes. If not defined or initiated properly, this can lead to large differences in the estimations. Hence, it should be noted that, although most of these models predict the annual global mean in a similar range, the geographic breakdown distribution of DMS (Fig. S3) can show large differences (Hulswar et al., 2022; Belviso et al., 2004b; Bock et al., 2021; Wang et al., 2020). The largest differences are seen in the Southern Ocean (Figs. 3 and S3). There is also a high spatial heterogeneity in the Southern Hemisphere (Figs. 1 and 2). This region has high productivity and high DMS emissions, which can have a large impact on aerosol formation, as compared to the Northern Hemisphere.

## 4  Summary and conclusions

In this study, we compared the latest interpolation-based and two parameterization-based seawater DMS estimations, which are used for calculating the sea–air fluxes of DMS in conjunction with a sea–air exchange parameterization. The interpolation-based method is easy to implement, but it results in a higher area-weighted global annual mean DMS (2.28 nM for H22) compared to other methods. The parameterization-based methods define a non-linear rela-

tionship between DMS and environmental and/or biogeochemical parameters through regression analysis and estimate lower weighted annual mean DMS compared to the interpolation-based method (1.69 nM for G18 and 1.75 nM for W20). W20 estimates a $\sim 3.4\%$ higher weighted global mean DMS when compared with G18, but it also shows a lot of geographical heterogeneity. In the case of the interpolation-based climatology (H22), the DMS estimate is biased towards regions where observations are frequently taken or towards the region of blooms. The method may give low or high DMS values depending on the sampling bias. For example, low DMS values are estimated in April in the northern polar region as compared to in March and May ($> 60° N$) (Fig. 3a). Thus, the interpolation method is not free from regional biases, particularly in the Arctic region.

The parameterization-based approaches depend heavily on the resolution of the proxy parameters, but there is a limitation regarding the satellite-data-based proxy parameterization. For example, in the Southern Ocean environment, due to presence of sea ice, satellite data do not generate robust PAR and thus are more restricted to the south of $50° S$ in early spring and late autumn, due to which the DMS climatology generated gets biased. G18 does not show peak values in the Southern Ocean during austral summer at a course resolution of $1° \times 1°$, but there is coastal enhancement at higher latitudes, and the method explains $50\%–57\%$ of the DMS variability compared to the observations, while W20 explains $66\%$ of the DMS variability. G18 shows lower values in the Southern Ocean compared to in the Northern Hemisphere. This low DMS in the Southern Ocean is one of the limitations of the G18 method.

Comparatively, W20 performs better than G18 in the Southern Hemisphere. However, not all blooms are resolved, which could be due to the global filtering (where in situ DMS $> 100$ nM is removed) before training the ANN model. The filtering of the response variable (DMS) and the predictors is probably done as the ANN model is sensitive to outlier points that could lead to overfitting of the model. McNabb and Tortell (2023) trained an ensemble ANN model in the South-

ern Ocean with DMS concentration values of more than 100 nM at a high resolution (20 km × 20 km), which is able to capture DMS hotspots in the Southern Ocean. Our observation from machine learning models suggest that machine-learning-based estimations have the potential to predict DMS accurately but need reliable high-resolution input data. These can also capture mesoscale variability, which is not possible with interpolation methods based on in situ observations directly. However, machine learning estimations need a large dataset across different biogeochemical provinces to train the models. Another machine learning model known as Gaussian process regression (GPR) was recently applied by Mansour et al. (2023); this was able to address $\sim 71\%$ of the DMS variability at high temporal (daily data) and spatial ($0.25° \times 0.25°$) resolutions in the North Atlantic Ocean for the prediction of DMS concentration. With fewer DMS points ($\sim 2236$), the model results show that this can be an efficient tool for obtaining seawater DMS concentration and that it may be successful in other oceanic regions or in the entire global ocean as well.

Finally, the interannual trends are calculated for the parameterization-based methods (G18 and W20), and a positive and significant trend ($t_b > 2$) in both G18 ($6.94 \pm 1.44\%$ per decade) and W20 ($3.53 \pm 0.53\%$ per decade) is obtained. This analysis using SeaWiFS data shows that there is an increase in DMS concentration over the period from 1998 to 2010. It is not possible to obtain past and future DMS projections from the satellite-based products as these products are available for a limited number of years; this could be solved through the parameters obtained from CMIP6 models, which are subject to quality-controlling and proper validation.

It should be noted that there is considerable uncertainty in the estimated DMS concentration and in the global distributions due to biases in the observations, unsuitable global filtering for all regions, incorrect interpolation, and the sensitivity of coefficients in parameterization methods. The area-weighted global annual means of G18 and W20 are within the range of biogeochemical models (1.51–1.98 nM), but the CMIP6 models do not necessarily show the same geographical breakdown distribution (Fig. S3) compared to H22. It should be noted that the climatologies show poor agreement in the Southern Hemisphere. This region is important due to its high productivity and, hence, high DMS concentrations and can have a large impact on aerosol formation compared to the Northern Hemisphere. The uncertainties in calculating seawater DMS concentrations can lead to large uncertainties in total DMS fluxes (please see Part B).

*Data availability.* All the data used here are publicly available, and links are provided in the paper.

*Supplement.* The supplement related to this article is available online at: https://doi.org/10.5194/bg-21-1-2024-supplement.

*Author contributions.* ASM conceptualized the study. SDJ analysed the data with help from SH. CAM, MG, TGB, and RS helped with the data, ideas, and understanding of the study. SDJ and ASM wrote the paper with the help of all the co-authors.

*Competing interests.* The contact author has declared that none of the authors has any competing interests.

ther geographical representation in this paper. While Copernicus Publications makes every effort to include appropriate place names, the final responsibility lies with the authors.

*Acknowledgements.* The Indian Institute of Tropical Meteorology is funded by the Ministry of Earth Sciences, Government of India. Martí Galí and Rafel Simo acknowledge support from the European Research Council (ERC) under the European Union's Horizon 2020 research and innovation programme (grant agreement no. 834162 – SUMMIT Advanced Grant to RS) and the Spanish Government through the grant GOOSE (no. PID2022_140872NB_I00), as well as the "Severo Ochoa Centre of Excellence" accreditation grant (no. CEX2019-000928-S).

*Financial support.* This research has been supported by the NAME OF FUNDER (grant no. GRANT AGREEMENT NO). TS11

*Review statement.* This paper was edited by Peter Landschützer and reviewed by two anonymous referees.

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

**Remarks from the language copy-editor**

CE1   Re-worded for clarity. Please confirm the changes and ensure that your meaning has remained intact.
CE2   Re-worded for clarity. Please confirm the changes and ensure that your meaning has remained intact.
CE3   Re-worded for clarity. Please confirm the changes and ensure that your meaning has remained intact.
CE4   Please confirm. Alternatively, should this be changed to "Kara Sea–Laptev Sea region"?
CE5   Please confirm. Alternatively, should this be changed to "Chorne Sea–Celtic Sea region"?
CE6   Please confirm the change and ensure that your meaning has remained intact.
CE7   Please confirm the changes and ensure that your meaning has remained intact.
CE8   Please check this table carefully for changes, as these might not appear in the track-changes file.
CE9   Please double-check and confirm capitalization (or lack thereof).
CE10  Please double-check and confirm capitalization (or lack thereof).
CE11  Please double-check and confirm capitalization (or lack thereof).

**Remarks from the typesetter**

TS1   Please provide Department.
TS2   Please provide Department.
TS3   Please provide Department.
TS4   Please send a new Supplement as a *.pdf without the title, authors, correspondence author, etc. as we will generate a Supplement title page during publication (with a citation including the DOI), which will contain this information.
TS5   Please check author names for consistency in the paper and the system: "Anoop S. Mahajan" or "Anoop Sharad Mahajan"? "Christa A. Marandino" or "Christa Marandino"? "Thomas G. Bell" or "Thomas Bell"?
TS6   Please confirm.
TS7   Please add Johnson, 2010 to the reference list.
TS8   Please provide your last access date.
TS9   Please provide your last access date.
TS10  Please provide your last access date.
TS11  Please note that there is funding information given in the acknowledgements, but you did not indicate any funding upon manuscript registration. Therefore, we were not able to complete the financial support statement. Please provide the missing information and double-check your acknowledgements to see whether repeated information can be removed from the acknowledgements. Thanks.
TS12  Please ensure that any data sets and software codes used in this work are properly cited in the text and included in this reference list. Thereby, please keep our reference style in mind, including creators, titles, publisher/repository, persistent identifier, and publication year. Regarding the publisher/repository, please add "[data set]" or "[code]" to the entry (e.g. Zenodo [code]).
TS13  Please provide article number or page range.
TS14  Not mentioned in the text.
TS15  Please provide article number or page range.
TS16  Please provide article number/page range and DOI number.
TS17  Please provide persistent identifier (DOI number preferred, url combined with last access date) and number of pages.
TS18  Please provide persistent identifier (DOI number preferred, url combined with last access date) and number of pages.
TS19  Please provide article number/page range and DOI number.
TS20  Please provide article number or page range.
TS21  Please provide article number or page range.
TS22  Please provide volume number.
TS23  Please provide article number or page range.
TS24  Please provide article number or page range.
TS25  Please provide article number or page range.