# Peer review of "Dimethyl sulfide (DMS) climatologies, fluxes and trends - Part A: Differences between seawater DMS estimations"

_EGUsphere, 2024_

## Author Comment (AC1)

**Response to Reviewer 2 comments for manuscript ID egusphere-2024-173. The comments are given in an italic typeface, and the responses are given in a bold typeface. The corresponding changes in the revised manuscript are highlighted in red.**

*2.1) General comments:*

*Joge et al examine the regional and temporal variability in DMS by comparing three climatologies, one of which was generated by interpolating observational data and two of which are parameterisation based. In addition, they generate the trend in DMS for a fourteen-year period using the two parameterisation-based climatologies.*

**Response: We thank the reviewer for thoroughly reviewing the manuscript. The responses to each specific comment are given in detail below.**

*2.2) Scientific significance: 3*

*The science question is important, as improved estimates of the oceans DMS source are required; however, the paper is somewhat brief and perfunctory, and not very "stretchy", as it just compares the outputs of the three different climatologies without testing sensitivities and only providing limited interpretation. In addition, the Conclusions could have been more substantial; for example, Figures 1 and 2 highlight that agreement between climatologies was poorest during summer in the southern hemisphere where marine DMS emissions will arguably have greatest impact on aerosol chemistry. This point could then have generated a recommendation in the Conclusion and abstract.*

**Response: We thank the reviewer for the comments regarding the analysis. We have included more details as per the comments. Regarding the Southern Ocean, the reviewer is right in highlighting the implications. The difference in the DMS sea-air fluxes in the Southern Ocean has been discussed in more detail in Part B of the manuscript. It can impact aerosol chemistry, which is also addressed in Part B of the study (https://egusphere.copernicus.org/preprints/2024/egusphere-2024-175/). We have, however, added this important point to the conclusions:** *'It should be noted that the climatologies show poor agreement in the Southern Hemisphere. This region is important in terms of high productivity and, hence, high DMS concentrations and can have a large impact on aerosol formation compared to the Northern Hemisphere region. The uncertainties in calculating seawater DMS concentrations can lead to large uncertainties in total DMS fluxes (please see Joge: Part B).'* **(Line 356-359)**

*2.3) Scientific quality: 2*

*Scientific methods and assumptions clearly outlined and description of experiments and calculations are sufficient to enable reproduction*

**Response: We thank the reviewer for the above comment.**

*2.4) Presentation Quality: 1*

*Presentation, referencing and language are all fine apart from a few typos*

**Response: We thank the reviewer for the above comment. Typos have been corrected throughout the manuscript.**

*Specific comments*

*2.5) Title - is a little misleading. "Fluxes" are mentioned in the Introduction but there is no generation or presentation of fluxes in the analysis.*
**Response: The complete work is divided into Part A and Part B with a common title, 'Dimethyl sulfide (DMS) climatologies, fluxes and trends', which connects Part A and Part B. Part A focuses on the Differences between seawater DMS estimations and Part B on the Sea-air fluxes.**

*2.6) Line 65 notes "are usually to the order of 0.25°×0.25° and hence can include mesoscale dynamic changes" yet Lines 97-99 identify that environmental parameters are at coarser resolution with SST, salinity & nutrients at 1°×1° and MLD at 0.5°×0.5°. Consequently, the DMS climatologies are generated at 1-degree resolution (Line 102), but how significant is this lower resolution in terms of the generated DMS and the differences between climatology outputs? Could these parameters be scaled to a higher resolution? Re-gridding is noted as a possible reason in Line 130 but not discussed and the reader is instead referred to G18.*
**Response: This line 65 was about current biogeochemical models, which can simulate processes to obtain seawater DMS concentration at 0.25° x 0.25° resolution. Hence, they can give mesoscale variability information. In theory, as the reviewer states, one could downscale the input parameters to higher resolution to estimate DMS at higher resolutions. Most global chemistry-climate models run at a much lower resolution, and hence even a 1x1 degree resolution is sufficient. Machine learning models can definitely help with this and have been used to get high-resolution regional DMS concentrations. But for a comparison between the methods, we believe it is better to stick to the lowest resolution so that the differences are not influenced by the downscaling of the proxy parameters.**

*2.7) Lines 129-144 My interpretation of Figure 1b is that there is generally poor agreement between the climatologies in the southern hemisphere and perhaps this point should be made clearer. The text says that "a band of elevated DMS is seen in the South Atlantic and Indian Oceans centered around the 45° S latitude as the satellite data of chlorophyll may be biased towards colored dissolved organic matter (CDOM) and detritus on the Argentinian basin." and this comment appears to be directed at G18. Why does the potential bias of CDOM & detritus on the Argentine basin only influence the G18 climatology, when satellite chlorophyll data were used in all climatologies? The band of elevated DMS at 45S is restricted meridionally in G18, whereas H22 and W20 show broader meridional spread – does this reflect interpolation in G18, or something else? Further analysis of this discrepancy would be useful.*
**Response: As suggested by the reviewer, new text is added in the updated manuscript: 'There is poor agreement between all three climatologies in the Southern hemisphere. A band of elevated DMS in the South Atlantic and Indian Oceans centered around the 45° S latitude is seen in G18 (Figure 1a). This is because chlorophyll a satellite data may be biased towards colored dissolved organic matter (CDOM) and detritus in the Argentinian basin (Astoreca et al., 2009; Hayashida et al., 2020; Bock et al., 2021). Thus, making chlorophyll a; a poor predictor by itself. This region is the transition between subtropical and subpolar**

*waters and is also known for high abundances of DMS produces like coccolithophores and dinoflagellates (Balch et al., 2016). However, H22 and W20 show a broader meridional spread (Fig. 1a). G18 which uses regression-based parameterization, and has coefficients sensitive to the PAR, and hence light absorbing fractions such as CDOM and detritus thus is most likely biased'* **(Line 142-149)**

*2.8) Fig 2A. It would be interesting to see this plotted as a proportional rather than an absolute difference in DMS.*

**Response: As requested, the new figure below is added in the supplementary text as Figure S2.**

[Figure]

**Figure S2: (a)** Proportional seasonal differences with respect to H22 climatology. **(b)** Total number of pixels in each bin of 20 % difference.

*2.9) Results section is quite detailed in description of regional differences between climatologies, but doesn't highlight the fundamental point that spatial disagreement is poorest in the southern hemisphere summer where DMS is arguably having a greater impact on aerosol formation than in the Northern hemisphere.*

**Response: We have now included the importance of the differences in the Southern Ocean in the results (*The largest differences are seen in the Southern Ocean (Fig.3 and S3). There is also a high spatial heterogeneity in the Southern Hemisphere (Fig.1 and 2). This region has high productivity and high DMS emissions, which can have a large impact on aerosol formation as compared to the Northern Hemisphere – Line 308-311) and also in the conclusions (*It should be noted that the climatologies show poor agreement in the Southern Hemisphere. This region is important in terms of high productivity and, hence, high DMS concentrations and can have a large impact on aerosol formation compared to the Northern**

*Hemisphere region. The uncertainties in calculating seawater DMS concentrations can lead to large uncertainties in total DMS fluxes (please see Joge: Part B)* – **Line 356-359). Part B also focuses on the flux differences, which are the largest in the Southern Ocean.**

*2.10) It's not clear in the Methods section how the limited availability of Southern Ocean environmental data is accounted for. For example, satellite data does not generate robust PAR data where sea ice is present, and the general availability of satellite data is restricted south of 50S in early spring and late autumn which may bias DMS climatologies that are reliant on satellite-derived environmental data. Again, this limitation could be discussed.*
**Response: This is an important point which has now been included as follows:** *It should be noted that there is a limitation for using satellite data as proxy data. For example, if we consider the Southern Ocean, satellite data does not provide robust PAR values where sea ice is present, and the general availability of satellite data is restricted south of 50º S in early spring and late autumn, which may bias the DMS climatology'* **(Line 112-115)**

*2.11) I was disappointed that there wasn't a further analysis of the sensitivity of, and so error derived from, factors such as interpolation (for example, by testing different interpolation approaches) and spatial resolution (comparing climatologies developed using different spatial scales and so potentially accommodating for mesoscale eddies).*
**Response: In H22 (Hulswar et al. 2022), the analysis related to the sensitivity of interpolation is already explained in terms of the Radius of Influence (ROI). On a global scale, each ROI resulted in a different global mean, but once the ROI dropped below 25 km, the mean value stabilized at ~2.44 nM. Although the global mean did not change by much, large regional differences were observed, with smaller ROI values showing less patchiness in the resultant climatology, indicating that choosing an appropriate ROI is crucial for an accurate estimation of the DMS distribution.**
**Most of the interpolation-based climatologies are developed at a coarse resolution (Hulswar et al. 2022; Lana et al. 2011). Climatologies developed using parameterization method based on machine learning are of much higher resolution (McNabb et al. 2023), which captures oceanographic features such as eddies, hydrographic fronts and jets that appear to play an important role in driving DMS variability in the Southern Ocean. In Wang et al. (2020) (W20), a sensitivity test is carried out on raw data (spatial resolution same as in situ DMS) and binned data in which model performance in terms of RMSE increases due to binning and averaging original dataset before training and hence the predictions of DMS concentrations.**
**In our analysis we focused on finding the spatial differences and why these differences occur among these published climatologies as it contributes to uncertainty in total fluxes in the atmosphere. We do not focus on the higher resolution data as it is not possible to get them in interpolation-based methods, unless we perform downscaling, which would introduce new uncertainties.**

*2.12) Line 229. Why is the trend not examined in H22?*
**Response: H22 is only a climatology and hence by definition there will be no trend.**

*2.13) Section 3.4 primarily describes (and repeats) Table 1, and so could be reduced*
**Response: We have simplified this section and presented it as bullet points as suggested by reviewer 1.**

*2.14) The Conclusions section is largely a Discussion and so should be divided into two sections.*
**Response: This section is renamed to *Summary and conclusions*. The main takeaway points are now divided into paragraphs.**

*Technical corrections*

*2.15) Line 84-85 Explain "the interconnected input, hidden and output layers"*
**Response: A short explanation is added: *'ANN is composed of layers of interconnected nodes. These nodes are organized into three layers: input layer, hidden layer and output layer. The hidden layer performs complex computations on the parameters obtained from the input layer and trains itself according to the parameters given to this layer. Once it is trained, the ANN becomes capable of predicting DMS values at a single node in the output layer.'* (Line 89-93)**

*2.16) Line 146 Where is the Corne Sea?*
**Response: This was a typo has been corrected to Chorne Sea. (At line 161)**

*2.17) Line 210 These are not "decreases" but instead are underestimates*
**Response: *'decreases'* replaced by *'underestimations'* (line 228 in modified manuscript)**

*2.18) Line 224 "shows" should be "showing"*
**Response: Corrected.**

*2.19) Line 255 "parametrization"*
**Response: *Parametrization* is also spelled as *Parameterization*. So, to keep uniformity *Parametrization* is replaced by *Parameterization* throughout the manuscript.**

*2.20) Line 291 missing word*
**Response: The text on a line 291 in old manuscript is re-written as : *The interpolation-based method is easy to implement but it results in higher area-weighted global annual mean DMS (2.28 nM for H22) compared to other methods'* (Line 314-316)**

*2.21) Line 295 "W20 estimates ~3.4 % higher weighted global mean DMS"*
**Response: *Changed.***

*2.22) Line 318 "there is an increase…."*
**Response: *'an'* added.**

---

## Author Comment (AC2)

**Response to Reviewer 1 comments for manuscript ID egusphere-2024-173. The comments are given in an italic typeface, and the responses are given in a bold typeface. The corresponding changes in the revised manuscript are highlighted in red.**

*1.1) General comments:*

*Sankirna et al. (2024) have presented an important evaluation of three of the most recent dimethylsulfide (DMS) climatologies, plus a few online model parameterisations, providing a timely guide to modellers who may be deciding on how DMS should be represented in their systems. This paper also provides an important benchmarking of the updated DMS climatology interpolated from observations (Hulswar et al. 2022), which may be expected to replace the most commonly used Lana et al. (2011) climatology. While this paper is not long, and does not provide analysis beyond statistical comparison, it addresses an important question clearly. I have only minor comments on this manuscript and would recommend its publication after they have been addressed.*

**Response: We thank the reviewer for the above comments and for identifying that this paper is a timely study that will be useful to the modelling community. The answers to the specific comments are given below.**

*Specific comments:*

*1.2) Line 13: 'Most models' – I think you should be more specific here – you are talking about atmospheric models that represent aerosol processes.*
**Response: This line has been updated to *'Most atmospheric models that represent aerosol processes ….'* (Line 13).**

*1.3) Line 26: '... the impact of DMS on the radiative budget are very sensitive to the estimate used' – I'd perhaps remove the word 'very', as even if it's a 100% increase, if its 1nm to 2nm I don't think that would have a 'very' large impact on the radiative balance… Until it has been shown what the impact on the radiative balance is, I'd temper this argument.*
**Response: Updated to*: '… the impact of DMS on the radiative budget is sensitive to the estimate used.'* (Line 26)**

*1.4) Line 35: Sentence beginning with 'Thus ...' - this sentence is a bit long and a little bit confusing. I think you need to make more clear the feedback that you are alluding to.*
**Response: The sentence is rephrased as: *'CCN contribute to the formation of clouds, increasing cloud albedo. Due to this, DMS emissions have the potential to decrease solar radiation at the ocean surface, resulting in negative feedback'.* (Line 35-36)**

*1.5) Line 37: I think you need to make mention of the comparatively large amount of literature indicating that the CLAW hypothesis likely is not plausible in the complexities of the real world (e.g., Quinn & Bates, 2011), but you can at the same time quantify its importance to the global energy balance (e.g., Fiddes et al. 2018).*
**Response: The following lines are now added as suggested: *'Past studies have shown that this feedback cycle is more complex than the original CLAW hypothesis (Quinn and Bates,***

*2011) However, it is undeniable that DMS affects the radiative budget on a global scale. For example, Fiddes et al. (2018) showed that the removal or enhancement of marine DMS can change the atmospheric radiative effect at the top of the atmosphere by 1.7 and -1.4 W m$^{-2}$, respectively. (Mahajan et al., 2015b) showed that the difference between model simulations with and without DMS can result in an aerosol radiative forcing difference of -.179 W m$^{-2}$, with the difference exceeding 20 W m$^{-2}$ in the Southern Ocean. Hence, there is a need to understand the DMS cycle within the context of uncertainties and biases of the climate models (Fossum et al., 2018; Fiddes et al., 2018).' (Line 37-44).*

*1.6) Line 74: Can you provide a reference here: 'A recent study...'*
**Response: Added** *(Galí et al., 2015). (Line 82)*

*1.7) Methods section: I was a little bit confused here, you are using three data sets that are publicly available, but your writing makes it sound like you have re-run some of this analysis? Perhaps you can revise your writing a little in this section to make clear that you are describing the data sets and not your own methods.*
**Response: Indeed, for the comparison between the climatologies, we are using published datasets. To calculate long-term trend, we had to re-calculate the datasets of W20 and G18. This has now been made clear** *'As only monthly climatologies of DMS are available from G18 and W20 public data, the models from these two papers were re-run to get monthly estimates of DMS from year 1998 to 2010 in order to calculate the trends of seawater DMS.'* **(Line 104-106)**

*1.8) Line 100: Can you clarify that the input parameters you are discussing are those that went into the G18/W20 parameterisations?*
**Response: Clarified** *as 'The parameters used for W20 and G18 are ...' (Line 106-107)*

*1.9) Line 103: Were G18 & W20 data sets available over the exact same time periods (1998-2010)? Can you explain a little bit here why this time period? (I think you do later, but would be good to have it upfront).*
**Response: This has now been added in the methods section too, as suggested (Line 121).**

*1.10) Line 130: In light of your results here, could you comment on how effective using chlorophyll as a proxy is?*
**Response: Chlorophyll is one of the primary predictors for DMS at it indicates presence of different types of phytoplankton. However, it is true that it is not an effective predictor by itself due to the complex biogeochemistry of DMS. We have added a sentence regarding this: '*Thus, making chlorophyll a; a poor predictor by itself*' (Line 145-146).**

*1.11) Line 142: I wonder how many observations the H22 data set has in these regions? How would that impact the results?*
**Response: There is a total, 43,002 observations used in G18, 89,569 in W20 and 872,427 in H22 globally. Out of these observations, 1,610 points are available in the region > 60$^{\circ}$ S for G18, 12,666 for W20 and 620,454 for H22. The estimations are obviously impacted**

by the number of observations and hence we conclude in this paper that we need more observations for increased accuracy. The below Figure S1 is now added to the supplementary text to show this clearly.

[Figure]

**Figure S1:** In situ DMS observations used in G18, W20 and H22.

*1.12) Line 209-214: Can you speculate on why these differences exist across methods? Is it due to poor data availability to train on? Or the quality of the inputs?*

**Response: The main reason is the availability of observations across different regions and season and the second the data quality. We have added this in the manuscript. '***The difference in the methods is driven by various factors. The sensitivity of methods to certain parameters (or observation bias in the case of H22) is the primary driver. However, the main reason for this is the availability of high-resolution observations across different regions and seasons and also the quality of the observations. In the future, more observations will help resolve some of these differences***.' (Line 204-207)**

*1.13) Line 221: I think 'with' should be 'while*
**Response: Replaced.**

*1.14) Section 3.3 Long term trend: I think this section needs to have an acknowledgement that 12 years is in fact not a long-term trend, certainly not enough to understand the full variability*

*of a system with respect to important climatic and oceanographic events (e.g., ENSO). I think it's still a valuable contribution and the trends to appear quite large, but I think it just needs to be recognised that this is still really quiet a short period! (And starts with one of the strongest El Nino's recorded – I don't know how ENSO might affect DMS, but I would be surprised if it didn't!).*

**Response: Yes, we agree and have included the following in the revised manuscript: '…** *Even though an increasing trend is obtained in G18 and W20, this period is not sufficient to understand the long-term variability of the Earth system and the DMS response to it.'* **(Line 263-264)**

*1.15) Line 230: 'We used monthly …' – I'm not sure why this is here, as you don't mention any results from these data sets?*

**Response: This line has been removed and section modified.**

*1.16) Line 247: What do you mean by 'predictors obtained from CMIP5 and CMIP6 reconstructed models? I'm not confident that 'this issue can be resolved' using climate model output – I think there are lot of issues in the CMIP6 models still around these processes, so I wouldn't really trust what they suggest (as you just said – CMIP5 and CMIP6 suggest opposite trends for DMS, so there is large uncertainty there still!).*

**Response: We have modified this section to remove the reference to the CIMP models in light of the comments from the reviewer. We have replaced this by: '***In theory, this could be addressed using the machine learning code and proxies from climate model projections, although this has large uncertainties too***'. (Line 265-266)**

*1.17) Line 255: this paragraph and those below is pretty dense – could you perhaps use dot points to describe each model so it flows a bit more clearly? Also – perhaps this would be better in a methods section? Also, can you describe the time period you are using here? And where did you get this data from.*

**Response: Bullet points are added before description of each model as suggested '…** *These models are described as follows: ….'* **This is a comparison of the area weighted means and hence does not read well in the methods section. The citation for each data is mentioned. (Section 3.4)**

*1.18) Line 285: I would really love to see this geographic breakdown and perhaps a similar analysis to what you did with the other three data sets (I know that this has been done to a degree already e.g., Bock et al. 2021, but it would be nice to have it all in the same place & in comparison to H22).*

**Response: The new figure below is now added in the supplementary text (Fig. S3). Text is added at line 307-308 in modified manuscript: '...** *the geographic breakdown distribution of DMS (Fig. S3) can show large differences …'*

[Figure]

**Figure S3:** Latitudinal means for each month of CMIP6 models described in section 3.4 along with H22 climatology.